# Constants of motion network

**M. F. Kasim & Y. H. Lim**
Machine Discovery Ltd.
Oxford, United Kingdom
{muhammad, yi.heng}@machine-discovery.com

## Abstract

The beauty of physics is that there is usually a conserved quantity in an always-changing system, known as the constant of motion. Finding the constant of motion is important in understanding the dynamics of the system, but typically requires mathematical proficiency and manual analytical work. In this paper, we present a neural network that can *simultaneously* learn the dynamics of the system and the constants of motion from data. By exploiting the discovered constants of motion, it can produce better predictions on dynamics and can work on a wider range of systems than Hamiltonian-based neural networks. In addition, the training progresses of our method can be used as an indication of the number of constants of motion in a system which could be useful in studying a novel physical system.

## 1 Introduction

Noether's theorem [1] states that if a system has a symmetry, then there's a constant of motion corresponding to it. As it turns out, constants of motion play a significant role in understanding the world around us and are ubiquitous in almost every aspect of physics. Among of the most prominent examples are the conservation of energy, the conservation of momentum, as well as the conservation of angular momentum.

Historically, constants of motion were discovered by doing analytical works from observational data or from mathematical descriptions of the systems' equations of motion. For example, the law of conservation of energy was proposed by Châtelet based on Newton's work on classical mechanics [2]. With the recent emergence of employing neural networks for scientific discovery and learning systems' behaviour from observational data [3, 4, 5, 6], naturally it raises a question, "*can we find constants of motion of dynamical systems from their data and exploit them to make a better prediction?*"

A large body of literature has been moving into this direction by learning the Hamiltonian [6, 7] or its variations [8, 9] of a system. However, as most of the previous works focus on Hamiltonian and its variations, they work well in conserving the Hamiltonian or energy. However, when the systems have other constants of motion, the Hamiltonian-based works fail to discover and exploit those quantities. Here we present the "Constant Of Motion nETwork" (COMET) that can discover constants of motion of a system and exploit them to make a better prediction. In contrast to Hamiltonian-based networks [6, 7, 8, 9], COMET is not constrained to Hamiltonian systems and its coordinate choice, making it generally applicable to a wider range of systems as shown in Table 1. In addition, we also found that the training progress of COMET can be used as an indication of how many independent constants of motion there are in the system (see section 6) which could be a valuable hint in studying a novel physical system. Our implementation and experiments can be found in the public domain[1].

---

[1]https://github.com/machine-discovery/comet/

|  | NODE [10] | HNN [6] | NSF [7] | LNN [8] | COMET |
|---|---|---|---|---|---|
| Conserves energy |  | ✓ | ✓ | ✓ | ✓ |
| Works on general coordinates | ✓ |  | ✓ | ✓ | ✓ |
| Works on dissipative systems | ✓ |  |  |  | ✓ |
| Conserves other quantities |  |  |  |  | ✓ |

Table 1: Summary of the methods comparison. The compared methods are neural ODE, Hamiltonian neural network, neural symplectic form, Lagrangian neural network, and COMET (ours).

## 2 Related works

The simplest way to learn the dynamics of systems with neural networks is by using neural ordinary differential equation (NODE) [10]. NODE takes the full states of the system and produces the dynamics of the systems, i.e. the time derivative of the states. The simulation can then be run by solving the ODE from the output of NODE. As there is no inductive bias in the NODE, they typically struggle to conserve quantities that are important in some systems' dynamics, such as energy.

Hamiltonian neural network (HNN) [6] is an attempt to solve this conservation problem by learning the Hamiltonian and calculate the state dynamics from the Hamiltonian. It has been shown that with HNN, one can conserve the energy and produce better motion prediction in a long time horizon. Due to its simplicity and the elegance of the idea, HNN has been applied on a wide range of tasks and neural network architectures [11, 12, 13], and even on dissipative systems by adding a dissipation term [14, 15]. HNN can also be combined with symplectic integrator [16, 17] to produce a better result from the trajectory observations.

Despite its ability to conserve the energy, HNN is limited by the requirement of using canonical coordinates instead of arbitrary coordinates. Works on Lagrangian neural networks [18, 8] solve this limitation by learning the Lagrangian. Other attempts use neural symplectic form to learn the coordinate-free representation of Hamiltonian [7] or Poisson neural network to learn the Poisson system [9]. However, those works are still limited to Hamiltonian or Poisson systems.

## 3 COMET: constants of motion network

We start by denoting a set of states in a system as $\mathbf{s} \in \mathbb{R}^{n_s}$ where $n_s$ is the number of states. States are the internal parameters of a system that completely determines its dynamics without external influence. For example, in a classical particle motion, the particle's position and velocity constitute the states of the system. Without external influence, the change of the states typically depends on the states itself, i.e. $d\mathbf{s}/dt = \dot{\mathbf{s}}(\mathbf{s})$.

A constant of motion is a quantity that is conserved over the time in the system, like energy. In some systems, such as integrable systems [19], there are other quantities other than energy that is conserved, for example, momentum or angular momentum. These constants of motion can typically be described as a function of the states of the system, so we denote it as $\mathbf{c}(\mathbf{s}) \in \mathbb{R}^{n_c}$ with $n_c$ is the number of constants of motion. As their quantity is constant throughout the motion, their time derivative must be 0, or $d\mathbf{c}/dt = 0$. By taking the dependency of $\mathbf{c}$ on $\mathbf{s}$, the condition on $\mathbf{c}$ can be written as

$$\frac{d\mathbf{c}}{dt} = \frac{\partial \mathbf{c}}{\partial \mathbf{s}} \dot{\mathbf{s}} = \mathbf{0}, \tag{1}$$

where $\partial \mathbf{c}/\partial \mathbf{s}$ is an $n_c \times n_s$ Jacobian matrix where each row of the matrix is the gradient of each constant of motion with respect to the states $\mathbf{s}$. The equation above means that the state dynamics $\dot{\mathbf{s}}$ must be perpendicular to the gradient of each constant of motion.

To design a deep learning architecture that can simultaneously learn the constants of motion and learn the dynamics that conserve the constant of motion, we define two functions that depends on the states that can be constructed with neural networks, $\dot{\mathbf{s}}_0(\mathbf{s})$ and $\mathbf{c}(\mathbf{s})$. The function $\dot{\mathbf{s}}_0 : \mathbb{R}^{n_s} \to \mathbb{R}^{n_s}$ is the initial guess of the rate of change of the states. The function $\mathbf{c} : \mathbb{R}^{n_s} \to \mathbb{R}^{n_c}$ is the function that computes the constants of motion of the system. To ensure the constants of motion are conserved as in equation 1, we compute the state dynamics by orthogonalizing the initial guess $\dot{\mathbf{s}}_0$ with respect to

the gradient of every constant of motion,

$$\dot{\mathbf{s}} = \text{ortho}\left(\dot{\mathbf{s}}_0, \{\nabla c_1, \nabla c_2, ..., \nabla c_{n_c}\}\right),\tag{2}$$

where $c_i$ is the $i$-th element of the constants of motion $\mathbf{c}$ and $\text{ortho}(\mathbf{a}, \mathcal{V})$ is an operation to orthogonalize the vector $\mathbf{a}$ to every vector in the set $\mathcal{V}$.

## 3.1 Orthogonalization process

One way to produce an orthogonal vector against a set of vectors is by using QR decomposition, i.e.

$$\mathbf{A} = (\nabla c_1, \nabla c_2, ..., \nabla c_{n_c}, \dot{\mathbf{s}}_0)$$
$$\mathbf{Q}, \mathbf{R} = \text{QR}(\mathbf{A})$$
$$\dot{\mathbf{s}} = \mathbf{Q}_{(\cdot, n_c)} \mathbf{R}_{(n_c, n_c)},\tag{3}$$

where $\mathbf{Q}_{(\cdot, n_c)}$ is the last column of the matrix $\mathbf{Q}$, and $\mathbf{R}_{(n_c, n_c)}$ is the element at the last row and last column of the matrix $\mathbf{R}$. The first row of the equations above shows a construction of a tall matrix $\mathbf{A} \in \mathbb{R}^{n_s \times (n_c + 1)}$ where the first $n_c$ columns are the gradient of the constants of motion and the last column is the initial guess of the states rate of change, $\dot{\mathbf{s}}_0$. QR decomposition is usually implemented using Householder transformation [20] which produces much smaller numerical error than the alternative Gram-Schmidt process [21] in practice.

The QR procedure above imposes a constraint that the number of constants of motion must be less than the number of states, i.e. $n_c < n_s$. This is in agreement with the maximum number of independent constants of motion in an integrable system is $n_c = n_s - 1$ [19]. By using QR decomposition, COMET will try to find $n_c$ independent constants of motion. Reasoning behind this is given in Appendix **??**.

## 3.2 Training loss function

We need to train the two trainable functions in COMET, $\dot{\mathbf{s}}_0(\mathbf{s})$ and $\mathbf{c}(\mathbf{s})$, so that the state dynamics $\dot{\mathbf{s}}$ from equation 3 match the dynamics from the observation or training data, $\hat{\dot{\mathbf{s}}}$. In order to train COMET, the loss function in this case is constructed as

$$\mathcal{L} = \left\|\dot{\mathbf{s}} - \hat{\dot{\mathbf{s}}}\right\|^2 + w_1 \left\|\dot{\mathbf{s}}_0 - \hat{\dot{\mathbf{s}}}\right\|^2 + w_2 \sum_{i=1}^{n_c} \left\|\nabla c_i \cdot \dot{\mathbf{s}}_0\right\|^2,\tag{4}$$

where $w_\cdot$ are the tunable regularization weights. The first term of the loss function is the standard $L_2$ error where the prediction must match the training data. The second term of the equation above is included to accelerate the training process by making the initial guess $\dot{\mathbf{s}}_0$ to be as close as possible to the actual value of the states' rate of change. The third term is an additional regularization to help the discovery of the constants of motion.

## 4 Learning constants of motion from data

To demonstrate the capability of COMET to simultaneously learn both the dynamics and the constants of motion, we tested it in a variety of cases. For all the cases in this section, the training data were generated by simulating the dynamics of the system from $t = 0$ to $t = 10$. From the simulations, we collected the states $\mathbf{s}$ as well as the states rate of change, $\hat{\dot{\mathbf{s}}}$, which were calculated analytically and were added a Gaussian noise with standard deviation $\sigma = 0.05$.

There are 6 simple experiments performed to demonstrate the capability of COMET: (1) mass-spring, (2) 2D pendulum, (3) damped pendulum, (4) two body, (5) nonlinear spring, and (6) Lotka-Volterra. The cases were selected to represent a wide variety of cases. It includes cases with Hamiltonian in canonical coordinates (case 1, 4, 5), Hamiltonian with non-canonical coordinates (case 2, 6), a case with redundant states (case 2), dissipative system (case 3), and a case with a moderate number of states (case 4). The details of each case, including the number of constants of motion that we set for COMET training, as well as the training setups are described in appendix **??**.

For each case, we compared the performance of COMET with other methods: (1) simple neural ODE (NODE) [10], (2) Hamiltonian neural network (HNN) [6] with the coordinates given in each case below, (3) neural symplectic form (NSF) [7], and (4) Lagrangian neural network (LNN) [8]. The neural network architecture for each method is detailed in appendix **??**.

| Case | NODE [10] | HNN [6] | NSF [7] | LNN [8] | COMET |
|---|---|---|---|---|---|
| mass-spring | $0.17^{+0.10}_{-0.13}$ | $0.19^{+0.24}_{-0.17}$ | $0.22^{+0.13}_{-0.17}$ | $\mathbf{0.12^{+0.08}_{-0.09}}$ | $\mathit{0.10^{+0.15}_{-0.09}}$ |
| 2D pendulum | $0.087^{+30}_{-0.067}$ | $0.10^{+13}_{-0.09}$ | $\mathit{0.11^{+0.24}_{-0.10}}$ | $\mathbf{0.029^{+0.29}_{-0.013}}$ | $0.18^{+0.17}_{-0.14}$ |
| damped pendulum | $\mathit{0.14^{+0.03}_{-0.05}}$ | $110^{+10}_{-110}$ | fail | fail | $\mathbf{0.007^{+0.014}_{-0.005}}$ |
| two body | $460^{+980}_{-460}$ | $\mathit{0.49^{+340}_{-0.33}}$ | fail | fail | $\mathbf{0.42^{+0.48}_{-0.39}}$ |
| nonlinear spring | $0.63^{+0.38}_{-0.35}$ | $\mathit{0.13^{+0.71}_{-0.11}}$ | $0.19^{+0.70}_{-0.15}$ | $0.17^{+0.70}_{-0.14}$ | $\mathbf{0.23^{+0.40}_{-0.18}}$ |
| Lotka-Volterra | $0.12^{+0.36}_{-0.10}$ | $0.65^{+1.6}_{-0.59}$ | $\mathit{0.080^{+0.20}_{-0.071}}$ | N/A | $\mathbf{0.048^{+0.055}_{-0.041}}$ |

Table 2: Root mean squared error of 100 randomly initialized simulations for each case and each method. The main number is the median while the range represents the 95% percentile (i.e. lower and upper bounds are 2.5% and 97.5% percentiles, respectively). The bolded values are the ones that give the best upper bound among other methods, while the italic values denote the second best. "fail" means that there are integration failures with scipy's `solve_ivp` which makes it unable to integrate to $t = 100$ in a reasonable time.

## 4.1 Results

For each case, we tested each method by running another 100 simulations from $t = 0$ to $t = 100$ (10 times longer than the training) using 1000 sampled points with the initial condition randomly initialized as above using different seed. The root mean squared errors of the state predictions are shown in table 2.

From table 2, we can see that COMET performs well across the test cases. In the mass-spring case, all methods perform well. However, when it goes to the pendulum cases and Lotka-Volterra, HNN fails to predict the dynamics due to the chosen coordinates not being the canonical coordinates. Although NSF can perform reasonably well in 2D pendulum, it fails on the damped pendulum case because it is not a Hamiltonian system. COMET takes the advantage of having constants of motion in the cases above that can be exploited to guide the true trajectory, therefore, it can achieve better predictions of the dynamics regardless of the chosen coordinates and whether it conserves energy or not.

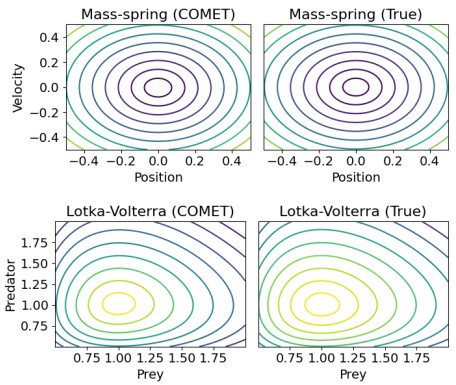

Figure 1: The contour plot of constant of motion discovered by COMET (left) compared to the true constant of motion (right) for mass-spring (top) and Lotka-Volterra (bottom) cases.

Figure 1 shows the discovered constant of motion in mass-spring and Lotka-Volterra cases. As it can be seen from the figure, COMET can successfully discovered the constants of motion from the data. Figure 2 shows the evolution of the known constants of motion for every method in the mass-spring, 2D pendulum, and the two body cases. The periodic variation from the true constants of motion are due to the added noise in the training data. In mass-spring case, figure 2(a) shows that the HNN, NSF, and COMET conserves the energy while the NODE gets the energy decreasing over time.

A different story can be found in figures 2(b) where they show 3 constants of motion of pendulum in 2D coordinate. In this case, NODE and HNN fail to conserve the constants of motion, while COMET can conserve those constants of motion during long period of time. The failure of HNN can be attributed to the state coordinates not being the canonical coordinates. This shows that COMET can discover the constants of motion with much less constraints in the state coordinates than HNN.

For the two body case in Figure 2(c), we can see that NODE and NSF diverges quite quickly. The failure of NSF in this case might be due to the added noise in the training data. Among the tested methods, only HNN and COMET conserves the energy. However, as HNN is only designed for Hamiltonian or energy conservation, it fails to conserve other quantities, such as the momentum and angular momentum. COMET, on the other hand, can successfully conserve those quantities.

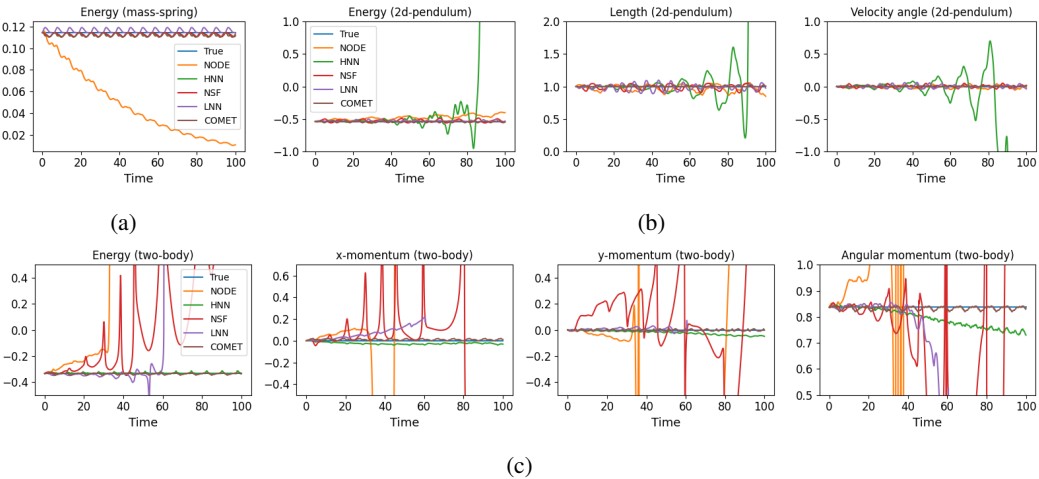

Figure 2: The constants of motion calculated for every method for (a) mass-spring, (b) 2D pendulum, and (c) two body. Please note that the integration of NSF and LNN for the two body case cannot be completed.

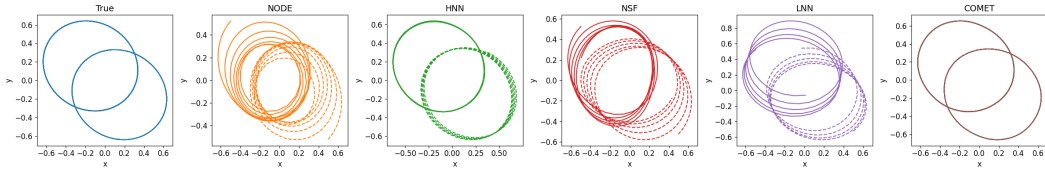

Figure 3: Motion trajectory of the simulated two body system using Neural ODE, HNN, NSF, and COMET (ours) from $t = 0$ to $t = 20$.

Figure 3 shows the trajectory of the two body system simulated using various methods tested here from $t = 0$ to $t = 20$. From the figure, we can see that only our method (COMET) that can produce closed trajectory. HNN produces almost closed trajectory, but it slightly deviates from the closed trajectory because it conserves only the energy, but does not necessarily conserve the other quantities. By exploiting as many constants of motion as possible, COMET can reproduce the motion with the small error compared to the other methods.

## 5   Systems with external influences

One advantage of COMET is that it can easily work with systems with external influences, such as external forces. If the system has conserved quantities when the external influences are kept constant, then COMET with a simple modification can be used to learn the constants of motion and exploit them to get more accurate dynamics. The modification is just to make the initial guess of the dynamics and the constants of motion to depend on the external influences as well as the states, i.e. $\dot{\mathbf{s}}_0(\mathbf{s}, \mathbf{x})$ and $\mathbf{c}(\mathbf{s}, \mathbf{x})$, where $\mathbf{x} \in \mathbb{R}^{n_x}$ is the external influence. The dynamics can still be calculated following the equation 2.

We conducted an experiment using the 2D pendulum from the section 4, but with additional external force in the $x$-direction, $F_x$. The training data was generated by having the external force with profile $F_x(t) = a_0 \cos(a_1 t + a_2)$ with uniformly-distributed random values of $a_0 \sim \mathcal{U}(-0.5, 0.5)$, $a_1 \sim \mathcal{U}(0, 5)$, and $a_2 \sim \mathcal{U}(0, 2\pi)$. The experiment was done similarly like in section 4, by adding the force as the input to the neural network for NODE, HNN, NSF, and LNN as well.

Figure 4 shows the constants of motion on the test system with constant external force. As seen on the figure, the values of the true constants of motion produced by COMET is oscillating slightly around a constant offset, due to the added noise in the training. In contrast, NODE produces the shift of the energy values, NSF produces large oscillation even for the energy, and LNN quickly diverges. Although HNN can produce similar energy deviation with COMET, it has larger deviation on other

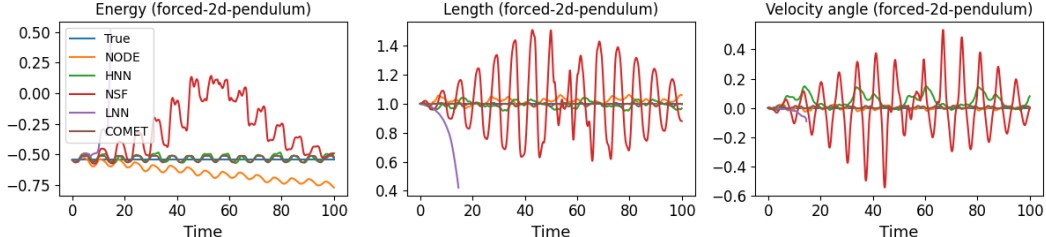

Figure 4: Constants of motion of the forced 2D pendulum case calculated using NODE, HNN, NSF, LNN, and COMET.

conserved quantities. This shows that even with external forces, COMET can still find and exploit the constants of motion for its dynamic predictions.

# 6 Finding the number of constants of motion

For most systems, the number of independent constants of motion is usually not known beforehand and not so obvious. Knowing the number of constants of motion can be useful in understanding the manifold dimension of the motion, however, this problem is not an easy problem to solve. During the research of this work, we observe that COMET's training progresses might provide a valuable indication to the number of constants of motion.

The parameter to look at is the first term in the loss function in equation 4, i.e

$$\mathcal{L}_1 = \left\| \dot{\mathbf{s}} - \hat{\dot{\mathbf{s}}} \right\|^2. \tag{5}$$

If we set the number of constants of motion greater than the true number, then that term could not get lower than a certain value. It is because $\dot{\mathbf{s}}$ is constrained to be perpendicular to the constants of motion and if there are excessive constants of motion, then it may not be able to match the value from experiments to a certain accuracy.

We ran a simple experiment to find the number of constants of motion for known systems. Specifically, COMET was trained in the damped pendulum, two body, and 2D nonlinear spring cases from section 4 without added noise and ran for 3000 epochs. Those cases are known to have 2, 7, and 2 constants of motion out of 4, 8, and 4 number of states, respectively. The numbers of constants of motion were scanned from 0 to the maximum value, $n_s - 1$. Figure 5 shows the value of $\mathcal{L}_1$ for the various cases with varying number of constants of motion.

From the figure 5 (top), we can see that once the number of constants of motion is set above a certain number, the value of $\mathcal{L}_1$ suddenly increases. This gives an indication of the actual number of constants of motion. If the system has the maximum number of constants of motion, then the values of $\mathcal{L}_1$ will always be similar to the values with $n_c = 0$. Besides the final value of $\mathcal{L}_1$, the evolution value of $\mathcal{L}_1$ for various number of constants of motion can be an indication on the true number of the constants of motion as we can see in figure 5 (bottom). From figure 5, we can see that the number of constants of motion for the damped pendulum case is 2, for the two body case is 7 (the maximum number), and for the nonlinear spring case is 2.

## 6.1 Failure mode

This technique in determining the number of constants of motion depends on the ability of the neural network to find the constants of motion. Therefore, if the neural network is not expressive enough, it could fail to find the constants of motion and indicate a lower number of constants of motion than it should be.

Figure 6 illustrates this case where we scanned the number of constants of motion from 0 to 3 in the 2D nonlinear spring case where the neural network only has 50 hidden elements per layer instead of 250. It gives an indication that the number of constants of motion to be 1 instead of the true number of 2.

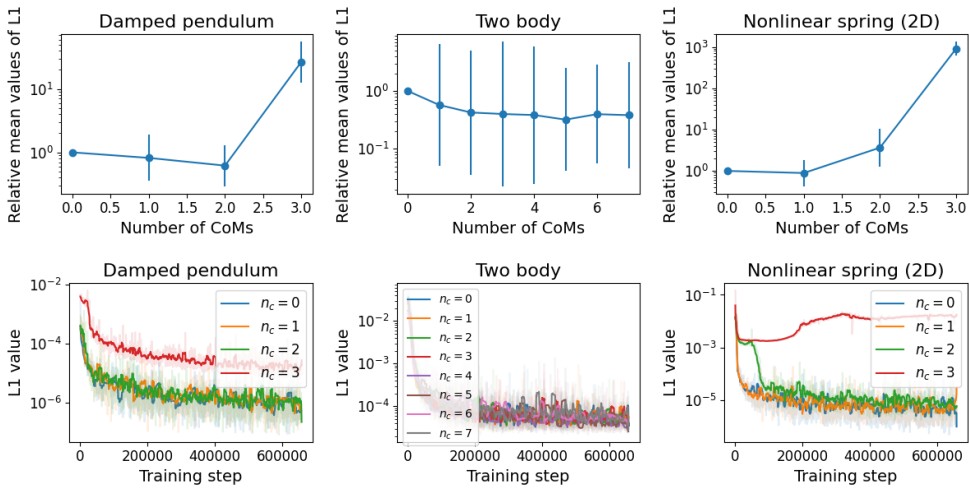

Figure 5: (Top row) The relative mean values of $\mathcal{L}_1$ from equation 5 for damped pendulum, two body, and nonlinear spring cases, with number of constants of motion $n_c$ were scanned from 0 to $n_s - 1$. The relative values were calculated by dividing it by the value of $\mathcal{L}_1$ at $n_c = 0$. The values and the error bars were respectively obtained by taking the mean and std from 5 COMETs trained with different random seeds. (Bottom row) The values of $\mathcal{L}_1$ during the training for various numbers of constants of motion for damped pendulum, two body, and nonlinear spring cases.

# 7 More complex cases

**Simulating a system with infinite number of states** — The previous examples only involve systems with finite or countable number of states. To demonstrate the general applicability of COMET, we ran an experiment on simulation of systems with infinite (but discretized) number of states. Specifically, we trained the COMET to learn the dynamics of shallow wave following Korteweg-De Vries (KdV) equation [22, 23] of $u(x,t)$, $\frac{\partial u}{\partial t} = -u\frac{\partial u}{\partial x} - \delta^2 \frac{\partial^3 u}{\partial x^3}$. The states in this case are the values of $u$ along the $x$-axis which constitutes infinite number of states.

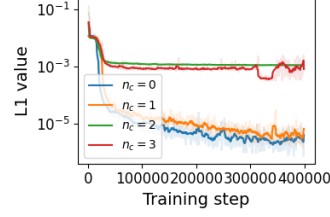

Figure 6: The failure of finding the number of constants of motion using a smaller network.

In our experiment, we simulate the behaviour of $u$ from $x = 0$ to $x = L = 5$ with periodic boundary condition, sampled in 100 points with uniform spacing. We also set $\delta = 0.00022$ for numerical stability. The training dataset was generated by running 100 simulations with random initial condition from $t = 0$ to $t = 10$ with 100 steps. The initial condition in the training dataset is $u(x,0) = a_0 + a_1 \cos(2\pi x/L + a_2)$ where $a_0$, $a_1$, and $a_2$ are randomly chosen within the range of $(1.5, 2.5)$, $(0, 1)$, and $(0, 2\pi)$, respectively.

The neural network was constructed with 1D convolutional layers with kernel size 5 and circular padding, followed by logsigmoid activation function. The pattern above was repeated 4 times but without the activation function for the last one, using 250 channels in the hidden layers. The training was done as described in section 4 which takes about 5-7 hours on an NVIDIA T4 GPU. The number of channels in the input is 1 (only for $u$), and for the output it is $1 + n_c$ where $n_c$ is the number of constants of motion that we set. The first channel of the output is to represent the initial guess of the dynamics, $\dot{u}_0(x)$. The last $n_c$ channels are for what we call as the constants of motion density, $p_i(x)$, for $i = 1, ..., n_c$. From $p_i(x)$, the constants of motion can be calculated as $c_i = \int_0^L p_i(x) \, dx$. Using the outputs from the network, the dynamics can be calculated following the equation 2.

We compared the performance of NODE and COMET in solving the KdV equation for $t = 0$ to 20. It is not obvious how to apply HNN, NSF, and LNN as the KdV equation has only $u$ as its states and do not include velocity nor momentum. Figure 7 shows the states $u(x,t)$ at $t = 20$ of the simulations using the true dynamics, NODE, and COMET. From the figure, we can see that at $t = 20$,

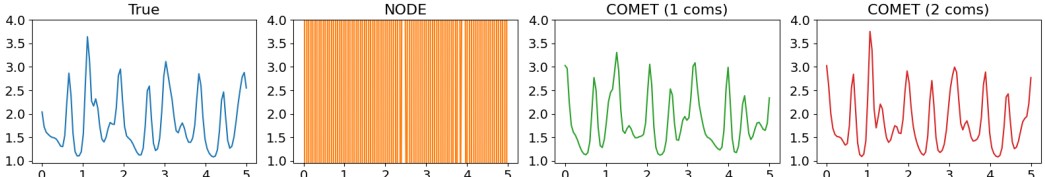

Figure 7: Plot of $u(x, t)$ at $t = 20$ from simulations done by the true analytic expression, NODE, and COMET using 1 and 2 constants of motion. All simulations were initialized to the same initial condition. The simulation run by NODE already diverges at $t = 20$.

the simulation done by NODE has diverged while COMET simulations are still intact. This shows that COMET can take advantage of the constants of motion to make its prediction more accurate.

**Learning from pixels** — Another experiment that we run to show COMET's capability is to learn the dynamics from pixels and to see if it can exploit the constants of motion in the latent space of an auto-encoder. In this case, we simulate the dynamics of two-body with gravitational interactions (like in section 4) and present the data as $30 \times 30$ pixels images that show the positions of the 2 objects.

Our model consists of an encoder, a COMET, and a decoder. The inputs to our model are 2 consecutive frames to capture the information about positions and velocities which we denote as $\mathbf{x}$. The encoder converts the pixel inputs to latent variables, $\mathbf{s} = \mathbf{f_e}(\mathbf{x})$. The dynamics of the latent variables are then learned by COMET which will then be converted back to pixel images, $\tilde{\mathbf{x}} = \mathbf{f_d}(\mathbf{s})$, by the decoder.

The loss function in this case is simply a sum of the COMET loss from equation **??** and the auto-encoder loss, i.e. $\mathcal{L}_{ae} = \|\mathbf{x} - \tilde{\mathbf{x}}\|^2$. The observed state derivatives $\hat{\dot{\mathbf{s}}}$ to compute the COMET loss in equation **??** is approximated by finite difference of the encoded pixel data from 2 consecutive time steps, i.e. $\hat{\dot{\mathbf{s}}} \approx [\mathbf{f_e}(\mathbf{x}(t + \Delta t)) - \mathbf{f_e}(\mathbf{x}(t))]/\Delta t$. In contrast to HNN and LNN, there is no requirement in COMET to make half of the latent states to be the time derivative of the other half.

In this experiment, we compared the performance of COMET, NODE, HNN, and NSF to learn the dynamics of the latent variables of the auto-encoder. The auto-encoder architectures are kept to be the same for all methods. The number of latent states are picked arbitrarily to be 10. It is more than the true number of states which is 8. For COMET, the number of constants of motion for COMET was found by following the procedure in section 6 which is 9.

Figure 8 shows a sample of the decoded images of the dynamics predicted by COMET, NODE, HNN, and NSF, compared to the ground truth. From the figure, we can see that the dynamics predicted by NODE diverges as soon as $t = 20$ and at some point the decoder produces non-sensible images. Although NSF and HNN can produce good images until the end of simulation at $t = 180$, the dynamics predicted by NSF and HNN diverges from the ground truth. In contrast, COMET can still match the dynamics of the ground truth simulation until the end of simulation. This shows COMET's capability in accurately learn the dynamics of latent states of an auto-encoder.

## 8 Discussions

**Limitations** – COMET works better if there is at least one constant of motion in the system. If there is no constant of motion, then COMET works similarly like the neural ODE [10]. Although we presented a way to find out the number of constants of motion in section 6, it still requires multiple training processes and manual insight.

In the case of successful training, COMET sometimes produces dynamics that are stiffer than the true dynamics, although LNN and NSF more often produce stiffer dynamics. In a rare case, the dynamics from COMET are so stiff that the integration by scipy's `solve_ivp` cannot be completed in a reasonable time. This only happens in the KdV case and did not happen in the other cases we tested for this paper. We believe that the limitations above should be addressed to move forward.

**Broader impact** – The impact of deep learning on physical sciences is expected to be similar to the impact of scientific computational method. Although it enables new fields of study, it adds more point of failure. For example, if a new or unexpected result is discovered using deep learning methods, it

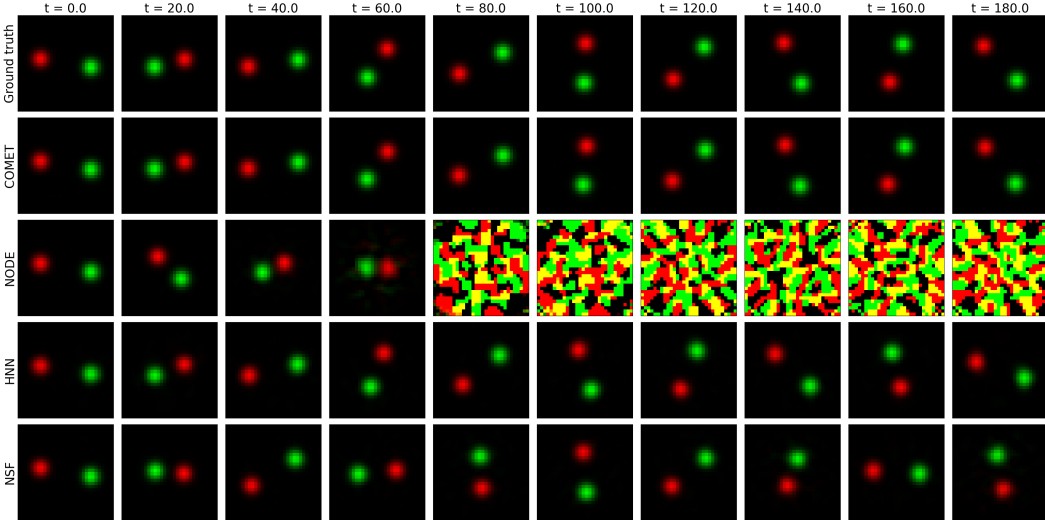

Figure 8: Snapshots of the decoded images of the dynamics predicted by COMET, NODE, HNN, and NSF, as well as the ground truth images.

could be a true discovery or false discovery due to the failure/imperfection in training, unsuitable neural network architecture, bug in the code, among other things. Therefore, deep learning methods such as COMET should be accompanied with other different and independent methods to confirm obtained results in scientific works.

## 9 Conclusions

We have shown that COMET can simultaneously learn the constants of motion and the dynamics of a system from observational data. Because the assumption made by COMET (i.e. have constants of motion) is less strict than Hamiltonian-based neural networks, it can be applied to a wider range of systems than the Hamiltonian-based neural networks, including dissipative systems and systems with external influences. The training progresses of COMET can also give an indication on the number of constants of motion in a system. With all the advantages we presented, we believe that COMET can be a valuable tool for scientific machine learning in the future.

## Acknowledgement

We would like to thank Sam Vinko and Brett Larder for their helpful comments in improving the manuscript as well as Ayesha Chairannisa for proofreading the paper.

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
