# Supplementary materials:
# Constants of motion network

**M. F. Kasim & Y. H. Lim**
Machine Discovery Ltd.
Oxford, United Kingdom
{muhammad, yi.heng}@machine-discovery.com

## A    More explanation on the training of COMET

The training loss of COMET is

$$\mathcal{L} = \left\| \dot{\mathbf{s}} - \hat{\dot{\mathbf{s}}} \right\|^2 + w_1 \left\| \dot{\mathbf{s}}_0 - \hat{\dot{\mathbf{s}}} \right\|^2 + w_2 \sum_{i=1}^{n_c} \left\| \nabla c_i \cdot \dot{\mathbf{s}}_0 \right\|^2. \tag{1}$$

With the terms above, it seems that the deep learning could cheat the training either by: (1) making all $c_i$ to be constants or (2) getting $\nabla c_i$ terms to be linearly dependent to each other, thus discovering fewer constants of motion than intended. However, in practice, we did not find both effects in the training. In fact, COMET training tends to find the constants of motion such that $\nabla c_i$ are all linearly independent although there is no explicit term to encourage linear-dependency of $\nabla c_i$ in the loss function.

This is due to the involvement of QR decomposition in computing the term $\dot{\mathbf{s}}$. For a QR decomposition of a matrix $\mathbf{A}$, i.e. $\mathbf{Q}, \mathbf{R} = \mathrm{QR}(\mathbf{A})$, the gradient of $\bar{\mathbf{A}} = \partial \mathcal{L}/\partial \mathbf{A}$ is

$$\bar{\mathbf{A}} = \left[ \bar{\mathbf{Q}} + \mathbf{Q}\mathrm{copyltu}(\mathbf{M}) \right] \mathbf{R}^{-T}, \tag{2}$$

where $\mathbf{M} = \mathbf{R}\bar{\mathbf{R}}^T - \bar{\mathbf{Q}}^T\mathbf{Q}$ and $\mathrm{copyltu}(\cdot)$ is the operation to copy the lower triangular elements to the upper triangular of the matrix. If the matrix $\mathbf{A}$ contains almost linearly dependent columns, the matrix $\mathbf{R}$ will be ill-conditioned. As the gradient above involves the inverse term of $\mathbf{R}$, the almost linearly dependent columns of $\mathbf{A}$ will produce very high gradient in $\bar{\mathbf{A}}$. This will make it harder to get the columns of matrix $\mathbf{A}$ (and therefore $\nabla c_i$) to be completely linearly dependent as it has to go through the region where the gradient is very high.

As the training of COMET tends to find independent constants of motion, setting the number of constants of motion higher than it should be would make the training fail. This is what we exploit in section 6 to give us the indication of the true number of constants of motion.

The last term in the loss function of COMET does not need the information from the dataset. Moreover, we would like the constraints to be fulfilled for the states outside the ones listed in the training dataset. Therefore, when training our COMET, the last term of the loss function is calculated using the states from the training plus some noise, i.e.

$$\mathcal{L} = \left\| \dot{\mathbf{s}}(\mathbf{s}) - \hat{\dot{\mathbf{s}}} \right\|^2 + w_1 \left\| \dot{\mathbf{s}}_0(\mathbf{s}) - \hat{\dot{\mathbf{s}}} \right\|^2 + w_2 \sum_{i=1}^{n_c} \left\| \nabla c_i(\mathbf{s} + \tilde{\mathbf{s}}) \cdot \dot{\mathbf{s}}_0(\mathbf{s} + \tilde{\mathbf{s}}) \right\|^2, \tag{3}$$

where $\tilde{\mathbf{s}} \sim \mathcal{N}(\mathbf{0}; \sigma^2 \mathbf{I})$ is the Gaussian noise with standard deviation $\sigma = 0.1$.

## B    Experiment details

This section contains the experiment details for cases tested in section 4. In section 4, there are 6 simple experiments performed to demonstrate the capability of COMET: (1) mass-spring, (2) 2D

pendulum, (3) damped pendulum, (4) two body, (5) nonlinear spring, and (6) Lotka-Volterra. The cases were selected to represent a wide variety of cases.

For each case, 100 simulations with random initial conditions were generated with 100 sampled time points each from $t = 0$ to $t = 10$. The dataset is split to 70% for training, 10% for validation, and 20% for test. The training was performed with batch size of 32 using a neural network with 3 hidden layers of 250 elements each using logsigmoid as the activation function to get infinitely differentiable function. There are $n_s$ inputs to the neural network with $n_s + n_c$ outputs. The first $n_s$ elements are assigned to the initial guess, $\dot{\mathbf{s}}_0$, while the next $n_c$ elements are assigned to the constants of motion, $\mathbf{c}$. The training procedures were performed using Adam optimizer [1] with the learning rate $3 \times 10^{-4}$ until 1,000 epochs, which takes about an hour with an NVIDIA T4 GPU. The regularization weights are $w_1 = w_2 = 1.0$.

The description for each simulation case can be found below.

**Case 1: Frictionless mass and spring.** This is the simplest case to test COMET's capability where an object of mass $m = 1$ is connected to a stationary point by a spring with constant $k = 1$. The states of this system is $\mathbf{s} = (x, \dot{x})^T$ where $x$ is the displacement of the object from its equilibrium position and $\dot{x}$ is the velocity of the object. The training data was generated by randomly initializing the position and velocity with a uniform random distribution between $(-0.5, 0.5)$. In this case, there is only one independent constant of motion which is energy, $E = (x^2 + \dot{x}^2)/2$.

**Case 2: 2D Pendulum.** The second case is a 2D pendulum of length $l = 1$ and mass $m = 1$ with an influence of gravity $g = 1$. The observed states in this case are the pendulum's $x$ and $y$ coordinate from the pivot as well as its velocity in $x$ and $y$ coordinate, i.e. $\mathbf{s} = (x, y, \dot{x}, \dot{y})^T$, making it redundant. The training data were generated by randomly initializing the angle and angular velocity with uniform distribution in the range $(-1.0, 1.0)$. There are three independent constants of motion in this case, (1) energy: $E = (\dot{x}^2 + \dot{y}^2)/2 + y$, (2) length: $x^2 + y^2 = 1$, and (3) angle: $x\dot{x} + y\dot{y} = 0$.

**Case 3: 2D damped pendulum.** This case is similar to the previous case, except that we introduced the damping force proportional to the velocity with damping coefficient $\alpha = 1$, making it an underdamped system. The training data were generated in a similar way as the previous case. As the energy is not conserved, only the second and third constants of motion from the previous case are valid.

**Case 4: Two body interactions.** We considered a case where two bodies of the same masses $m = 1$ are interacting with gravitational force with constant $G = 1$ and rotating around their centre of mass. The training data were generated by initializing it with a distance randomly chosen between $(1.0, 3.0)$ with perpendicular velocity between $0.7v_0$ to $1.0v_0$ where $v_0$ is the velocity to make the orbits circular. As the motion is planar, we only considered their motion on a 2D plane. Therefore, there are 8 state variables, $\mathbf{s} = (x_1, y_1, x_2, y_2, \dot{x}_1, \dot{y}_1, \dot{x}_2, \dot{y}_2)^T$. As the two-body motion is well-known to be fully integrable, the number of constants of motion is $n_s - 1$, which equals to 7. Among them are: total energy, total angular momentum, and total $x$ and $y$ momentum.

**Case 5: 2D nonlinear spring.** We consider a case of a motion of an object of mass $m = 1$ in 2D where it is connected to the origin with a nonlinear spring with force $\mathbf{F} = -|\mathbf{r}|^2\mathbf{r}$ where $\mathbf{r}$ is the position of the object in 2D coordinate. The states in this case is $\mathbf{s} = (x, y, \dot{x}, \dot{y})^T$. The dataset was generated by starting the simulation with randomly selected states between $(-1.0, 1.0)$ for all positions and velocities. The constants of motion of this systems are the energy and the angular momentum, which makes $n_c = 2$.

**Case 6: Lotka-Volterra** equation is an ordinary differential equation modelling the population of predator and prey. It is known to have a symplectic structure [2], therefore it has a constant of motion. We consider the equations $\dot{x} = x - xy$ and $\dot{y} = -y + xy$ where $x$ and $y$ represent the prey and predator populations respectively. There are only 2 states here, $\mathbf{s} = (x, y)^T$ with $n_c = 1$. The initial values of $x$ and $y$ are sampled randomly from uniform distribution within $(0.5, 2.0)$. As there is no time derivative variable in the states, it does not make sense to apply LNN for this case.

## B.1 Learning from pixels

In the learning from pixels in section 7, we generated the data using the dynamics of the two-body case described above. Each image was generated based on the location of the two-body in the simulation.

The neural networks consist of encoder, decoder, and the dynamics learner. The encoder and decoder are multi-layer perceptron (MLP) with 3 hidden layers where there are 256 hidden elements each. Each linear layer in the encoder and decoder is followed by log-sigmoid activation function, except for the last layer. Following [3], the weights in the MLP were initialized to be orthogonal. For the dynamics learner, we use the different architectures based on each tested method. The details of the architecture can be found in appendix C.

The training loss in this case was composed of the reconstruction loss and the dynamics loss. As opposed to [3], we do not have an auxiliary loss to force half of the states to be derivatives of the other half. If the image pixels are contained in a vector $\mathbf{p}$, the loss can be written as

$$\mathcal{L} = \|\lambda \mathbf{f_{dec}}\left(\mathbf{f_{enc}}\left(\mathbf{p}\right)\right) - \mathbf{p}\|^2 + \mathcal{L}_{dyn} \tag{4}$$

where $\mathbf{f_{dec}}(\cdot)$ and $\mathbf{f_{enc}}(\cdot)$ are the decoder and encoder functions, respectively, $\lambda$ is the relative weight of the reconstruction loss, and $\mathcal{L}_{dyn}$ is the dynamics loss. The dynamics loss for COMET follows the equation (5). Following the work in [3], we use $\lambda = 10$.

## C  Neural network architectures

In this section, we will explain in more detail about the architecture of the tested methods in section 4 as well as in section 5 in incorporating external forces. We use the same notations as in section 5, where the state is $\mathbf{s} \in \mathbb{R}^{n_s}$, its time derivative is $\dot{\mathbf{s}} \in \mathbb{R}^{n_s}$, the external force is $\mathbf{x} \in \mathbb{R}^{n_x}$, $n_s$ is the number of states, and $n_x$ is the number of external forces.

The neural networks for all methods tested here (including COMET) consists of 3 hidden layers with 250 elements each with logsigmoid activation function. The activation function is only applied to the hidden layers, but not applied to the output layer. However, the number of inputs and outputs might be different, depending on the needs of each method. The neural network architecture above is chosen to produce good training results for all methods.

### C.1  Neural ODE

With neural ODE [4], the states' time derivative is directly represented by a neural network that takes the states, $\mathbf{s}$, as its input, i.e.

$$\dot{\mathbf{s}} = \mathbf{f_{NODE}}(\mathbf{s}). \tag{5}$$

The neural network takes $n_s$ inputs, produces $n_s$ outputs, with hidden layers and the activation function follow the description above.

If external forces, $\mathbf{x}$, exist, the neural network is modified to take $\mathbf{x}$ as the input as well, i.e.

$$\dot{\mathbf{s}} = \mathbf{g_{NODE}}(\mathbf{s}, \mathbf{x}). \tag{6}$$

In this case, the neural network architecture stays the same, but with concatenated $\mathbf{s}$ and $\mathbf{x}$ as its input, therefore taking $n_s + n_x$ inputs.

### C.2  Hamiltonian neural network (HNN)

HNN [3] was implemented by having a neural network takes $n_s$ input for $\mathbf{s}$ and 1 output for the predicted Hamiltonian, $H$,

$$H = f_{HNN}(\mathbf{s}). \tag{7}$$

For cases with canonical coordinate, the first half of the states represents the position while the last half of the states represents the canonical momentum. The time derivative of states is calculated by

$$\dot{\mathbf{s}} = \begin{pmatrix} \mathbf{0} & \mathbf{I} \\ -\mathbf{I} & \mathbf{0} \end{pmatrix} \frac{\partial H}{\partial \mathbf{s}} \tag{8}$$

The hidden layers and the activation function of the neural network follow the implementation in other cases.

If there are external forces, then the neural network is modified to take the concatenated $\mathbf{s}$ and $\mathbf{x}$ as its input, i.e.

$$H = g_{HNN}(\mathbf{s}, \mathbf{x}). \tag{9}$$

The time derivative of states is still calculated according to equation 8.

## C.3 Neural Symplectic Form (NSF)

NSF [3] was implemented by having 2 neural networks where both take the states $\mathbf{s}$ as the input. One neural network produces one output, $H$, that represents the Hamiltonian, and the other one produces $n_s$ outputs, $\mathbf{Y}$, for its skew-symmetric matrix. Mathematically, it can be written as

$$H = f_{NSF1}(\mathbf{s}) \tag{10}$$
$$\mathbf{Y} = \mathbf{f_{NSF2}}(\mathbf{s}). \tag{11}$$

Both neural networks have 3 hidden layers with 250 elements each with logsigmoid as their activation function. This follows the neural network implementation in other cases. Combining both functions above into a single neural network produces worse training results than having two neural networks. The time derivative of the states is obtained by calculating

$$\dot{\mathbf{s}} = \left[ \frac{\partial \mathbf{Y}}{\partial \mathbf{s}} - \left( \frac{\partial \mathbf{Y}}{\partial \mathbf{s}} \right)^T \right]^{-1} \frac{\partial H}{\partial \mathbf{s}}. \tag{12}$$

Similar to the other methods with external force, the neural networks are modified to take the concatenated $\mathbf{s}$ and $\mathbf{x}$ as their inputs, i.e.

$$H = g_{NSF1}(\mathbf{s}, \mathbf{x}) \tag{13}$$
$$\mathbf{Y} = \mathbf{g_{NSF2}}(\mathbf{s}, \mathbf{x}). \tag{14}$$

The time derivative of the states is still calculated according to equation 12 above.

## C.4 Lagrangian neural network (LNN)

The neural network in LNN [5] takes $n_s$ inputs (for $\mathbf{s}$) and produces 1 output for the predicted Lagrangian, $L$,

$$L = f_{LNN}(\mathbf{s}). \tag{15}$$

For cases with position and velocity as the states, the states are arranged so that the first half of the states is the position and the last half is the velocity, i.e. $\mathbf{s} = \left( \mathbf{q}^T, \dot{\mathbf{q}}^T \right)^T$. The time derivative of the states is given by $\dot{\mathbf{s}} = \left( \dot{\mathbf{q}}^T, \ddot{\mathbf{q}}^T \right)^T$ where the acceleration $\ddot{\mathbf{q}}$ is calculated by

$$\ddot{\mathbf{q}} = \left( \frac{\partial^2 L}{\partial \dot{\mathbf{q}} \partial \dot{\mathbf{q}}} \right)^{-1} \left[ \left( \frac{\partial L}{\partial \mathbf{q}} \right) - \left( \frac{\partial^2 L}{\partial \mathbf{q} \partial \dot{\mathbf{q}}} \right) \dot{\mathbf{q}} \right]. \tag{16}$$

With external force, the neural network takes the concatenated $\mathbf{s}$ and $\mathbf{x}$ as its input,

$$L = g_{LNN}(\mathbf{s}, \mathbf{x}), \tag{17}$$

where the acceleration still following the equation 16 above.

# D   Additional experimental results

## D.1   Partial number of constants of motion

Although the number of constants of motion can be found with the procedure in section 6, it can produce a number lower than the true number of constants of motion. Therefore, it is interesting to see the effect of setting lower number of constants of motion to COMET's predictions accuracy. Table 1 shows the error range of the COMET's predictions with varying number of constants of motion.

From the table, it can be seen that setting the number of constants of motion at least 1 could give better worst case predictions than at least one of the other tested methods (NODE, HNN, LNN, and NSF) in most cases. Increasing the number of constants of motion also decreases the 95th percentile bound of the prediction error, which means that it can keep the stability of the trajectory better as the number of constants of motion increases.

| Number of coms | 2D pendulum | damped pendulum | two body | nonlinear spring |
|---|---|---|---|---|
| 1 | $0.69^{+7.8}_{-0.60}$ | $0.019^{+0.031}_{-0.009}$ | $0.79^{+3000}_{-0.33}$ | $0.37^{+0.52}_{-0.31}$ |
| 2 | $1.3^{+1.6}_{-1.1}$ | $\mathbf{0.007^{+0.014}_{-0.005}}$ | $\mathbf{0.64^{+0.36}_{-0.39}}$ | $\mathbf{0.23^{+0.40}_{-0.18}}$ |
| Full | $0.18^{+0.17}_{-0.14}$ | $\mathbf{0.007^{+0.014}_{-0.005}}$ | $\mathbf{0.42^{+0.48}_{-0.39}}$ | $\mathbf{0.23^{+0.40}_{-0.18}}$ |

Table 1: Root mean squared error of 100 randomly initialized simulations for COMET with varying number of constants of motion. The main number is the median while the range represents the 95% percentile (i.e. lower and upper bounds are 2.5% and 97.5% percentiles, respectively). The **bolded** values are the ones that give the better upper bound compared to all other non-COMET methods tested in this paper (NODE, HNN, LNN, and NSF), while the *italic* values are the ones that give the better upper bound than at least one of other non-COMET methods. "Full" means that it uses the true number of constants of motion described in section B. Some cases are excluded because they only have 1 constant of motion.