# OpenReview forum: "Constants of motion network"
_NeurIPS.cc/2022/Conference — NeurIPS 2022 Accept_

### Official Review · Reviewer_Jrbh · 2022-07-10

**Rating:** 7
**Confidence:** 3
**Soundness:** 3 good
**Presentation:** 3 good
**Contribution:** 2 fair

**Summary:**

In the context of learning models for dynamical systems, the authors propose to learn the constants of motion of the system by means of a neural mapping from states. The method is validated on several (albeit small) simulated dynamical systems, where it appears to improve upon previous approaches.

**Questions:**

The assumed number of constants of motion (n_c) appears to have a significant effect (Fig. 5). Maybe I missed it, but I couldn't find how these were selected in the benchmark examples?

It is a bit concerning that it degrades for 50 hidden neurons on the 2D non-linear spring system, with three layers there are still thousands of parameters in that network. Any idea why? Could this have something to do with biases induced by the QR decomposition? It is not obvious to me what kind of biases that will induce.


**Limitations:**

Mostly adequate.

**Strengths And Weaknesses:**

This paper was mostly clearly written (if occasionally abstract) and enjoyable to read. While mechanics is not my area of expertise, it proposes what seems like a novel idea for modelling of dynamical systems (learning constants of motion). It also empirically improves upon the results of the famous neural ODE paper and later developments. On the other hand, the evaluation is entirely on what seems like quite simple examples.

Concerns:
-----------------
Presentation:
- Some intuition of how the learning interacts with the integration scheme would have been helpful (c.f. NODE paper). It is quite abstract.

Experiments:
- The assumed number of constants of motion appears to have a significant effect (Fig. 5). How were these selected in the benchmark examples?

- It is a bit concerning that it degrades for 50 hidden neurons on the 2D non-linear spring system, there are still thousands of parameters in that network. Any idea why? Could this have something to do with biases induced by the QR decomposition? It is not obvious to me what kind of biases that will induce.

- As an engineer, I would have liked to see some example on how this approach can be used on some more complex real-world-like sysid problem.

Minor presentation issues:

"grad" -> writing out "gradient" in text would read better, or typeset it as a math operator (or nabla)

l282: works similar/is comparable? to

l290: which field, or just on science?
l290: more point*s*

---

> ### Author Response · Authors · 2022-08-02
> **Response to reviewer**
>
> Thank you for your detailed review.
>
> > The assumed number of constants of motion appears to have a significant effect (Fig. 5). How were these selected in the benchmark examples?
>
> The cases in section 4 are well-known physical systems where the number of constants of motion is known.
> For cases where the number of constants of motion is unknown, one can follow the procedure in section 6 to get the maximum number of constants of motion where COMET still works well.
> It can also be combined with other method to approximate the number of constants of motion, e.g. [Chen, *et al.*, 2022](https://doi.org/10.1038/s43588-022-00281-6), followed by the procedure in section 6 to find the number of constants of motion faster.
> Once the maximum number of constants of motion is found, one should use that in training COMET to provide better long-term predictions.
>
> > It is a bit concerning that it degrades for 50 hidden neurons on the 2D non-linear spring system, there are still thousands of parameters in that network. Any idea why? Could this have something to do with biases induced by the QR decomposition? It is not obvious to me what kind of biases that will induce.
>
> Our first guess is because it does not have sufficient expression capability to express the constants of motion.
> One of the constants of motion in the 2D non-linear spring system is the energy where it's proportional to $|\mathbf{r}|^4$ which might be hard to represent with smaller neural networks.
> Another thing is that training with QR tends to repel the gradients of constants of motion, $\nabla \mathbf{c}$, to be linearly independent to each other (it induces large gradient when the gradients are almost linearly dependent).
> This might make the training path more complicated to reach the correct constants of motion.
> We can't say for sure what's the problem, but those are our hypothesis.
> Proving it right or wrong and finding the right explanation is currently out of the scope of this paper and we will leave it out as a future work.
>
> > As an engineer, I would have liked to see some example on how this approach can be used on some more complex real-world-like sysid problem.
>
> Thank you for the pointer. We would also be interested to see this, however, as the time is limited, we currently cannot present it in the first round of rebuttal.
> If the referee could give some reference for the interesting sysid problem that is easy to setup, we would greatly appreciate it.
>
> **Minor presentation issues:**
>
> > "grad" -> writing out "gradient" in text would read better, or typeset it as a math operator (or nabla)
>
> Thank you, we've changed it in the revised version.
>
> > l282: works similar/is comparable? to
>
> With no constant of motion, the dynamics is just the direct output of the neural network without any postprocessing (i.e. QR in COMET).
> So we say "works similarly" instead of "is comparable".
>
> > l290: which field, or just on science? l290: more points
>
> Thank you, we've changed it to "deep learning on physical sciences", as it is the field that we're comfortable mentioning.

---

> > ### Comment · Reviewer_Jrbh · 2022-08-08
> > **Raised my score.**
> >
> > I thank the authors for their informative replies. Ideally I would have liked some more analysis of the QR part, but the paper is otherwise well written, the technical approach appears well founded, and it improves upon relevant baselines.

---

> > > ### Author Response · Authors · 2022-08-09
> > > **Thank you**
> > >
> > > We thank the reviewer for their reviews and for increasing the score.

---

### Official Review · Reviewer_FmUK · 2022-07-11

**Rating:** 3
**Confidence:** 4
**Soundness:** 2 fair
**Presentation:** 3 good
**Contribution:** 1 poor

**Summary:**

The paper is well written and easy to follow. It describes a methodology to learn a system of differential equations from data. The methodology can be used to capture the evolution of a given multi-dimensional time series s(t); possibly the time series can be influenced by an exogenous input (e.g. a force acting on the system). The working hypothesis is that the provided time series is the result of a constrained differential equation, i.e. there exists a function c(.) such that c(s) is constant on the trajectory s(t).

The proposed approach is tested on motion data (i.e. frictionless mass-spring, 2D pendulum, 2D damped pendulum, two body interactions, 2D nonlinear spring, Lotka-Volterra dynamics). Results show that the proposed method performs better than previous approaches (i.e. NODE, HNN, NSF, LNN) in terms of mean squared error on prediction. Some heuristics for choosing the number of constraints acting on the system are proposed; generalization to systems with infinite number of states is also proposed.

**Questions:**

- ***How robust is the proposed methodology to the choice of the system state?*** Do the authors have evidence that the proposed approach could potentially work replacing the given system state with an embedding (such as the latent of a VAEs) learnt from observations which are connected to the state but possibly higher dimensional (e.g. an image of the pendulum)?

- ***How robust is the proposed methodology to noise on the system derivative?*** The system state is always composed of the system generalized coordinates and their derivatives. As a result the system state derivative often contains the generalized coordinates accelerations (i.e. second order derivatives) which are extremely noisy when measured on real systems. Authors mentioned the states rate change was subject to a Gaussian noise with standard deviation of 0.05. Can authors elaborate on how this is representative of the typical noise observed in real systems when second order derivatives are approximated (e.g. finite differences)?

- ***What is the role of $s_0(s)$ at test time?*** Looking at (4) and (2), it seems that the only role of $s_0$ is (at training time) to guide $s$ to $\hat{s}$ and to be sure that at convergence $\dot{s} \in \langle \nabla c_1, \dots, \nabla c_n \rangle^\perp$. After training it is unclear whether the function $s_0$ is needed. Can author describe better the need to explicitly learn the function $s_0(s)$?

**Limitations:**

Not applicable.

**Strengths And Weaknesses:**

The paper is well written and easy to read. The structure of the paper is sound and results have been presented with enough details to understand the main point of the submitted paper. The major but fundamental weakness of the paper is that it focuses on relatively simple and artificial examples (e.g. mass-spring, pendulum) which aren't of practical use but merely useful for assessing the soundness of the approach. Even though I can see the value of these examples when explaining Lagrangian or Hamiltonian physics (where we need to understand how the different analytical components interact), I see very limited value in using these artificial examples when explaining how to use neural-networks to approximate a given time series. By focusing on this artificial examples, authors fall short in addressing fundamental questions listed below.

(1) ***Choosing the system state***. In classical mechanics  (e.g. Newton-Euler, Legrangian, Hamiltonian) the state of the system and its derived quantities (e.g. its derivative, the momentum) are fundamental quantities with specific properties. The choice of the state isn't trivial and it requires an analytical understanding of the system itself. In the proposed paper, authors assume that the state is given and often coincides with the one used in classical mechanics. However, in applications relevant to machine learning (as the one proposed), assuming that the state is given is quite a restrictive assumption.

(2) ***Choosing the system state dimensionality and number of constraints***. Section 6 proposes an heuristic for finding the number of constraints of motion. The problem is quite fundamental and it deserves a more thorough investigation. Authors could try to get inspiration from solutions to a similar problem in the scope of system identification and model selection (e.g. MDL, minimum description language for model selection; AIC, Akaike information criteria).

---

> ### Author Response · Authors · 2022-08-02
> **A new experiment with auto-encoder is added**
>
> Thank you very much for your thorough review.
>
> > (1) **Choosing the system state.** In classical mechanics the state of the system and its derived quantities are fundamental quantities with specific properties. The choice of the state isn't trivial and it requires an analytical understanding of the system itself. In the proposed paper, authors assume that the state is given and often coincides with the one used in classical mechanics. However, in applications relevant to machine learning (as the one proposed), assuming that the state is given is quite a restrictive assumption.
>
> One of the main point that we want to demonstrate in the paper is that COMET has less restrictions in terms of choosing the states compared to other methods, such as LNN and HNN.
> We showed this in the pendulum cases by choosing $x, y, \dot{x}, \dot{y}$ as the state variables instead of the usual $\theta, \dot{\theta}$ for HNN and LNN.
> Choosing 4 variables instead of 2 also shows that COMET can work with redundant state variables.
> We also showed this in the Lotka-Volterra case where we choose the simplest variables, not special canonical coordinates (i.e. the ones required by HNN), and not states that consists of $q,\dot{q}$ (i.e. the ones required by LNN).
> However, by having only a few cases for the illustration, we can see why this point is missed.
>
> To strengthen this point, we have added a new experiment in learning the two-body dynamics via images (2nd part in section 7 in the revised version).
> In this case, the images are encoded to latent state variables, and then decoded back to the images (i.e. auto-encoder).
> The dynamics of the latent state variables (which are learned and not given explicitly in the dataset) were learned succesfully by COMET.
> It was done even without additional constraints on the latent state variables (unlike HNN, for example).
> The success of this experiment shows that COMET can work even if the states are not given explicitly.
>
> > **(2) Choosing the system state dimensionality and number of constraints.** Section 6 proposes an heuristic for finding the number of constraints of motion. The problem is quite fundamental and it deserves a more thorough investigation. Authors could try to get inspiration from solutions to a similar problem in the scope of system identification and model selection.
>
> The main purpose of this paper is to learn the dynamics more accurately by exploiting the presence of constants of motion.
> It is a proposed improvement to HNN, LNN, NSF, as well as other works in terms of generalizability.
> Finding the number of constants of motion is also an interesting research field, but this is not the main focus of our paper.
> We present it as a means of maximizing the benefit of COMET for unknown systems by setting the number of constants of motion as close as possible to the true number.
> It can, of course, be combined with other methods (such as [Chen, *et al.*, 2022](https://doi.org/10.1038/s43588-022-00281-6)) to approximate the number of constants of motion, followed by the procedure from section 6 to speed up the process.
>
> **Questions**
>
> > How robust is the proposed methodology to the choice of the system state?
>
> COMET does not need a special set of state coordinates, unlike HNN (that requires canonical coordinates) and LNN (that requires the states to be consisted of $q,\dot{q}$).
> As long as the chosen state is sufficient to determine the dynamics, it can be used in COMET.
> To strenghten our point, we have added a new experiment in the revised version about learning the dynamics of a system in the latent space of an auto-encoder (following the reviewer's recommendation) without additional constraints on the latent variables.
> This shows that COMET is robust with the choice of the system state.
>
> > How robust is the proposed methodology to noise on the system derivative?
>
> The noise affects the accuracy of the trained model, but it does not affect much of the stability of the predicted trajectory as COMET tries to conserve a preset number of constants of motion.
> For example, adding noise to the two body case would make the orbit's frequency prediction a bit off, but the predicted trajectory will still be similar to the true trajectory (e.g. see Figure 3).
> On the other hand, methods such as HNN, LNN, and NSF only conserves one constant of motion.
> Hence, adding noise to those method affects the stability of the predicted trajectory more severely than COMET (e.g. Figure 3).
>
> > What is the role of $\dot{s_0}(s)$ at test time?
>
> $\dot{s_0}(s)$ is one of the direct outputs of the neural network besides $c(s)$, while $\dot{s}(s)$ is not the direct output of the NN.
> $\dot{s_0}(s)$ is used as the initial guess of the directions where it will be orthogonalized w.r.t. $\nabla c$ with QR decomposition to get $\dot{s}(s)$.
> In (4), the term $|\dot{s_0}-\hat{\dot{s}}|^2$ is to make the initial guess as close as possible to the true value, thus makes the training faster.

---

### Official Review · Reviewer_yjkt · 2022-07-12

**Rating:** 7
**Confidence:** 3
**Soundness:** 3 good
**Presentation:** 3 good
**Contribution:** 3 good

**Summary:**

In this paper, the authors proposed a novel method to solve dynamical systems through data. Instead of formulating the system Lagrangians or Hamiltonians, the method leverages the symmetry and conservation laws, thus does not assume any specific dynamical system forms. The proposed network predicts both the guess of the generalized velocity and the set of constants of the motions (COMs). Then, by exploring the orthogonality between the gradients of COMs and the generalized velocity, the final system velocity is projected through a QR decomposition.  The authors show that their method performs equally well or better than many baselines in many simulated dynamical systems.


**Questions:**

Besides the main comment, I have a few questions and suggestions for the authors:

1, Explain a bit more on why QR decomposition is chosen as the way to orthogonalization. What other methods are available? Why this particular one (i.e. differentiability, computation simplicity, etc)?
2, Explain a bit more on the “stiffness” of the dynamics. Why COMET, LNN, NST can produce stiffer dynamics?

3, Figure 5 shows that COMET works even without all COMs included. Can the authors add an ablation study to show how well COMET performs with a partial set of COMs?

**Strengths And Weaknesses:**

The strengths of the paper:

The method is quite novel yet simple, i.e. utilizes the system’s symmetry structure and conservation laws to better learn the system dynamics.

The proposed method can handle many different types of dynamical systems, including Hamiltonian systems, dissipative systems, and even infinite dimension PDEs.

The method outperforms many baselines in a wide range of tasks, and the generated trajectory has higher quality in some tasks (i.e. two body systems).


The method works well even with a partial list of constants of motions, as shown in Figure 5.


The weakness of the paper:

Most of the experiments are conducted in relatively simple dynamical systems. It will be interesting to compare the method with Lagrangian and Hamiltonian based methods in higher dimensional mechanical systems (such as robots).  I suspect that LNN may outperform the proposed method in some of these cases.

In table 1, LNN is shown to not work with dissipative systems, which I am not convinced of.  It will be surprising that a Lagrangian based method cannot handle dissipations.

The proposed method, COMET, performs significantly worse than other baselines such as LNN in the 2D pendulum task, and even worse than NODE, which is surprising–According to the authors, if the system does not have any symmetry it will be the same as NODE.

---

> ### Author Response · Authors · 2022-08-02
> **First response**
>
> Thank you for your elaborate review.
>
> > Most of the experiments are conducted in relatively simple dynamical systems.
>
> We presented systems that readers can easily understand so we can focus on demonstrating the wide applicability of COMET.
> One of the case we use in the paper is the two-body system where it has 8 states and section 7 contains an example of a PDE where it has infinite degrees of freedom.
> However, in order to include more complicated examples, we have added a new experiment in section 7 in the revised version about learning the dynamics of latent variables of an auto-encoder ("Learning from pixels").
>
> > It will be interesting to compare the method with Lagrangian and Hamiltonian based methods in higher dimensional mechanical systems (such as robots). I suspect that LNN may outperform the proposed method in some of these cases.
>
> We agree that LNN might outperform COMET in some cases. This is also apparent in our results in Table 1.
> Our hypothesis is that because LNN explicitly assumes that half of the states are the time derivative of the other half of the states, it contains the second-order bias that is shown by [Gruver, *et al.*, 2022](https://arxiv.org/pdf/2202.04836.pdf) to improve results.
> On the other hand, the assumption makes LNN does not work on cases where the assumption is invalid (e.g. KdV and Lotka-Volterra).
> Moreover, in some cases, we found that the dynamics learned by LNN is highly stiff that makes it cannot be integrated.
>
> > In table 1, LNN is shown to not work with dissipative systems, which I am not convinced of. It will be surprising that a Lagrangian based method cannot handle dissipations.
>
> The dynamics of LNN (i.e. equation 6 of [Cranmer, *et al.*, 2020](https://arxiv.org/pdf/2003.04630.pdf)) is derived by assuming the Lagrangian does not depend explicitly on the time $t$.
> However, for dissipative systems, the Lagrangian has an explicit dependence on time (e.g. see [Kobe, *et al.*, 1986](https://doi.org/10.1119/1.14840)) which violates the LNN assumption.
>
> > The proposed method, COMET, performs significantly worse than other baselines such as LNN in the 2D pendulum task, and even worse than NODE, which is surprising–According to the authors, if the system does not have any symmetry it will be the same as NODE.
>
> LNN performs better than COMET in those cases because it has the second-order bias as mentioned in our response above.
> However in some cases, LNN fails to integrate as it produces dynamics that are too stiff.
>
> When compared to NODE, the median error produced by COMET is slightly higher than NODE only in 1 out of 6 tested cases in section 4.
> However, the upper bound error of COMET is consistently lower than NODE in all of the tested cases, and sometimes it is significantly lower.
>
> **Questions answered**
>
> > 1, Explain a bit more on why QR decomposition is chosen as the way to orthogonalization. What other methods are available? Why this particular one (i.e. differentiability, computation simplicity, etc)?
>
> There are several algorithms to perform orthogonalization, such as: (1) Gram-Schmidt (GS) decomposition, (2) Householder (HH) transformation, or (3) Givens rotation (GR).
> GS is well-known for its severe numerical instability.
> HH provides a much better numerical stability and is also usually implemented in the modern QR decomposition, makes it easy to use.
> This is what we use.
> Another one, GR, provides an advantage of being easily parallelized. However, at this stage, parallelization on orthogonalization is not our top priority.
>
> > 2, Explain a bit more on the "stiffness" of the dynamics. Why COMET, LNN, NST can produce stiffer dynamics?
>
> We could not say for sure the reason of the stiffness of the methods, especially for LNN and NSF, but we have a hypothesis for COMET.
>
> Stiffness usually happens when some of the eigenvalues of Jacobian $\partial\dot{\mathbf{s}}/\partial\mathbf{s}$ are very high.
> As COMET uses QR decomposition, the Jacobian matrix depends on gradient of the QR, which depends on the inverse of $\mathbf{R}$ matrix (i.e. one of the QR outputs).
> If the matrix $\mathbf{R}$ is nearly singular, it will provide high gradient values that could make the dynamics stiff.
> The matrix $\mathbf{R}$ can be nearly singular if the training produces $\nabla c$ that are nearly linearly-dependent to each other.
> This case, although rare, it is possible to happen.
>
> > 3, Figure 5 shows that COMET works even without all COMs included. Can the authors add an ablation study to show how well COMET performs with a partial set of COMs?
>
> We have added a simple ablation study on comparing the predictions of COMET with varying number of constants of motion (currently only 1, 2, and the full constants of motion, but hopefully we can add more).
> We see that by increasing the number of constants of motion, the upper bound of the predictions error decreases, which means that it can keep the stability of the trajectory better as the number of constants of motion increases.

---

### Meta-Review · Area_Chair_xTFM · 2022-08-28

**Recommendation:** Accept
**Confidence:** Less certain

**Metareview:**

2 of the 3 reviewers highly appreciated the rebuttal and are now recommending the paper for acceptance without any reservations. The 3rd, most critical reviewer, FmUK did unfortunately not react. The new experiments "learning from pixels" nicely addresses the reviewer's concern about having to carefully choose the system state. Also the concern about the number of constants of motion is well addressed. The question about the sensitivity to noise could have been stronger: FmUK was talking about physical systems that typically have accelerations as part of their states, but typical sensors only measure positions/angles (which typically already produce slightly noisy measurements). Applying numerical differentiation twice to get to the accelerations often results in very noisy measurements. Hence the question how representative the experiments (where the states - that also include $\dot{x}$ - are assumed to be measured perfectly) are for real systems. I think this is still an interesting point to discuss - but no deal-breaker for me.

**Award:**

No

---

### Decision · Program_Chairs · 2022-09-14

Accept